# Does TBC1D4 (AS160) or TBC1D1 Deficiency Affect the Expression of Fatty Acid Handling Proteins in the Adipocytes Differentiated from Human Adipose-Derived Mesenchymal Stem Cells (ADMSCs) Obtained from Subcutaneous and Visceral Fat Depots?

**DOI:** 10.3390/cells10061515

**Published:** 2021-06-16

**Authors:** Agnieszka Mikłosz, Bartłomiej Łukaszuk, Elżbieta Supruniuk, Kamil Grubczak, Marcin Moniuszko, Barbara Choromańska, Piotr Myśliwiec, Adrian Chabowski

**Affiliations:** 1Department of Physiology, Medical University of Bialystok, Mickiewicza 2C Street, 15-222 Bialystok, Poland; bartlomiej.lukaszuk@umb.edu.pl (B.Ł.); elzbieta.supruniuk@umb.edu.pl (E.S.); adrian.chabowski@umb.edu.pl (A.C.); 2Department of Regenerative Medicine and Immune Regulation, Medical University of Bialystok, Waszyngtona 13 Street, 15-269 Bialystok, Poland; kamil.grubczak@umb.edu.pl (K.G.); marcin.moniuszko@umb.edu.pl (M.M.); 3Department of General and Endocrine Surgery, Medical University of Bialystok, M. Sklodowskiej-Curie 24a Street, 15-276 Bialystok, Poland; barbara.choromanska@umb.edu.pl (B.C.); piotr.mysliwiec@umb.edu.pl (P.M.)

**Keywords:** ADMSCs, adipocytes, CD36/SR-B2, FABPpm, FATP1, FATP4, subcutaneous and visceral fat depots, TBC1D4 (AS160), TBC1D1, fatty acid transporters translocation

## Abstract

TBC1D4 (AS160) and TBC1D1 are Rab GTPase-activating proteins that play a key role in the regulation of glucose and possibly the transport of long chain fatty acids (LCFAs) into muscle and fat cells. Knockdown (KD) of *TBC1D4* increased CD36/SR-B2 and FABPpm protein expressions in L6 myotubes, whereas in murine cardiomyocytes, TBC1D4 deficiency led to a redistribution of CD36/SR-B2 to the sarcolemma. In our study, we investigated the previously unexplored role of both Rab-GAPs in LCFAs uptake in human adipocytes differentiated from the ADMSCs of subcutaneous and visceral adipose tissue origin. To this end we performed a single- and double-knockdown of the proteins (TBC1D1 and TBC1D4). Herein, we provide evidence that AS160 mediates fatty acid entry into the adipocytes derived from ADMSCs. TBC1D4 KD resulted in quite a few alterations to the cellular phenotype, the most obvious of which was the shift of the CD36/SR-B2 transport protein to the plasma membrane. The above translated into an increased uptake of saturated long-chain fatty acid. Interestingly, we observed a tissue-specific pattern, with more pronounced changes present in the adipocytes derived from subADMSCs. Altogether, our data show that in human adipocytes, TBC1D4, but not TBC1D1, deficiency increases LCFAs transport via CD36/SR-B2 translocation.

## 1. Introduction

Adipose tissue had long been considered a quiescent fat-storage site; however, recent research into adipocytes, especially of white adipose tissue (WAT) origin, has altered that perception [1]. The tissue is now recognised as a complex and dynamic endocrine organ with an intricate role in whole-body homeostasis [2]. In humans, WAT may be divided into two major fat depots, i.e., subcutaneous (SAT) and visceral fat (VAT). The quantity of the latter of the two is positively correlated with an increased health burden, while SAT is instead considered an energy storage space and safety buffer against the negative consequences of obesity [3]. Consequently, the anatomical amount and distribution of various fat deposits influence the development of diseases coexisting with obesity. Adipose-tissue-derived mesenchymal stem cells (ADMSCs) are retrieved from fat depots and are a part of the so-called stromal vascular fraction (SVF) [2]. Currently, much attention is being paid to ADMSCs, which, in contrast to bone-marrow-derived MSCs (mesenchymal stem cells), are more numerous and are readily accessible using minimally invasive surgical procedures. This makes ADMSCs an important alternative source of stem cells suitable for clinical applications. Recent studies demonstrated ADMSCs’ potential as a treatment for metabolic complications of obesity, i.e., hyperglycemia, hyperinsulinemia, insulin resistance, T2DM, dyslipidemia and cardiovascular complications [4,5,6].

In the current study, we explored the biological role of the two Rab-GTPase activating proteins (RabGAPs) vitally involved in adipocyte substrate utilisation, namely TBC1D1 and/or TBC1D4 [7,8,9]. TBC1D4 (also known as Akt substrate of 160 kDa, hence AS160) is a major manager of insulin-dependent GLUT4 (glucose transporter type 4) trafficking to the plasma membrane, as confirmed by studies conducted on skeletal muscle and adipocyte tissue [10,11]. Knockdown of AS160 results in an increased basal GLUT4 plasma-membrane content in adipocytes [12]. Interestingly, aside from its role in GLUT4 translocation, recent studies suggested that TBC1D4 may also regulate the trafficking of fatty acid transport proteins [8,13]. Nowadays, it is known that several proteins are involved in the cellular uptake of long chain fatty acids (LCFAs), i.e., scavenger receptor B2 (CD36/SR-B2), plasma-membrane-associated fatty acid-binding protein (FABPpm) and fatty acid transport proteins (FATPs) [14,15]. It is unknown whether a deficiency in TBC1D1 or TBC1D4 affects adipocyte fatty acid uptake, and if so, whether it is solely caused by a change in the amount of fatty acids handling proteins or also by directing the FA transporters to the plasma membrane. Samovski et al. demonstrated that knockdown of AS160 redistributes CD36/SR-B2 to the plasmalemma in HL-1 murine cardiomyocytes [8]. This was caused by the decreased inhibitory influence of AS160, due to its silencing, on Rab8a protein. Previously, we demonstrated that, in L6 myotubes, TBC1D4 deficiency increased the expression of CD36/SR-B2 and FABPpm, which, in turn, enhanced fatty acid influx into the muscle cells [13]. Importantly, AS160 (TBC1D4) has a less known structural homolog, TBC1D1, which has also been implicated in GLUT4 incorporation into the plasma membrane [16]. Given the above, we decided to add two additional experimental groups to our study, namely, the cells with the knocked down *TBC1D1* and those with the simultaneous knockdown of both *TBC1D1* and *TBC1D4*. That experimental model should enable better discernment of the role of each protein. Additionally, a possible compensatory effect of one functional homolog for the lack of the other should not go unnoticed. Furthermore, that setup should allow for the control of a possible interaction between the two proteins.

Literature data indicate that both Rab-GAP proteins (TBC1D1 and TBC1D4) are likely to be signalling hubs of glucose and lipid metabolism [16,17]. However, investigations in the area have focused almost exclusively on muscle tissue [7,17]. Herein, we sought to expand the current body of research on adipocytes, i.e., the cells constituting another metabolically important tissue. We decided upon the cells differentiated in vitro from ADMSCs of subcutaneous and visceral fat origin. The tissue was obtained from lean female patients. Since adipocyte tissue can adapt to changes in the energy balance through the processes of hyperplasia or hypertrophy, differences in the fatty acid storage properties between mature adipocytes may well exist [18]. On the contrary, the cells differentiated from ADMSCs constitute quite a uniform population, and the expected response pattern to applied treatments should be more consistent. This confers significant advantages over primary adipocytes because they are relatively hard to cultivate and impose difficulties in preparing reliable materials for successive experiments [19].

To investigate the role of both Rab-GAPs in LCFAs uptake, we performed single siRNA-mediated knockdowns of either *TBC1D1* or *TBC1D4*, as well as combined knockdown of both RabGTPase-activating proteins. Detailed analysis encompassed evaluation of the expression of fatty acid handling proteins (CD36/SR-B2, FABPpm, FATP1, FATP4) at the transcript (mRNA) and protein levels. Importantly, the latter was assessed in the total cell homogenate as well as in the plasma-membrane fraction. This is of vital importance, since studies demonstrate that many interventions result in stable total fatty acid transporter content, with their cellular redistribution observed in the background [20,21]. As additional support for the insight coming from the before-mentioned procedures, we applied the assessment of radioactive palmitate uptake. This should allow us to determine the overall net fatty acid influx in case of possible opposite changes in the transporters’ expression (upregulation of some transporter levels and downregulation of others). All of the examinations were done in adipocytes differentiated from ADMSCs of both subcutaneous and visceral origin.

## 2. Materials and Methods

### 2.1. Patients

White adipose tissue (WAT) was isolated from the subcutaneous (abdominal region) and visceral (omental region) depots from 4 lean females (BMI < 25 kg/m^2^; postmenopausal women from 54 to 62 years of age) undergoing elective laparoscopic cholecystectomy. Visceral adipose tissue was collected using a harmonic knife for haemostasis. Burnt edges of visceral adipose tissue fragment were cut off with a surgical scalpel. Subcutaneous adipose tissue was collected using window forceps. The participants underwent clinical examination, anthropometric measurements and appropriate laboratory tests (Appendix A). Patients with acute inflammatory changes and a history of malignancy were excluded from the study. Patients were treated at the First Department of General and Endocrine Surgery at the University Hospital in Białystok. All procedures were designed, conducted, and reported in compliance with the Declaration of Helsinki and were approved by the Ethics Committee of the Medical University of Białystok (permission R-I-002/187/2017). Moreover, the Guidelines for Good Clinical Practice were upheld. All patients gave written informed consent that they were willing to take part in this study.

### 2.2. ADMSCs Isolation, Culture and Adipogenic Differentiation

The ADMSCs that were isolated from subcutaneous (abdominal region) fat and visceral (omental region) of adipose tissues of female donors underwent elective laparoscopic cholecystectomy. Briefly, adipose tissue was washed three times in phosphate-buffered saline (PBS, PAN-Biotech, Aidenbach, Germany) and then minced and digested in collagenase (250 U/mL collagenase NB 4G Proved Grade, Serva, Heidelberg, Germany) for 1 h at 37 °C. Afterwards, in order to separate the mature adipocytes from the stromal vascular fraction, the digest was filtered through a 500 µm strainer and centrifuged for 10 min at 600× *g*. The pelleted cells forming the stromal vascular fraction of the adipose tissue were resuspended in erythrocyte lysis buffer (Thermo Fisher Scientific, Waltham, MA, USA), filtered through a 200 µm and then 20 µm strainers and centrifuged again for 5 min at 600× *g*. The ADMSC-rich pellet was suspended in chemically defined medium designed for the optimal growth of mesenchymal stem cells in vitro—MSC medium containing growth supplements (MSCM, ScienCell Research Laboratories, Carlsbad, CA, USA), 5% fetal bovine serum (FBS, Thermo Fisher Scientific, Waltham, MA, USA) and 1% penicillin/streptomycin solution—and cultured. The viability of freshly isolated ADMSCs was measured via Trypan blue staining. Cell purity was assessed by flow cytometry as described below. The culture medium was changed every 2–3 days and a confluence of 80–90% cells were subcultured up to passage 2 or 3 (in order to reach the number of cells necessary to perform different assays). Thereafter, the cells were detached with trypLE (Thermo Fisher Scientific), centrifuged and diluted in freezing medium optimised for stem cell cryopreservation (Stem-Cellbanker DMSO Free, Takara Bio, Mountain View, CA, USA). In 1 mL of freezing media, 5 × 10^5^–5 × 10^6^ cells were collected in cryovials and placed directly in a −80 °C freezer overnight. The next day frozen vials were transferred to a liquid nitrogen storage tank. Cells from passages 3 or 4 were used for the experiments.

Frozen ADMSCs were thawed and, on average, 20,000 cells were plated onto a 6-well plate in MSC medium. Confluent cells (90–100% of confluency) were induced into adipocytes using manufacturer’s differentiation medium containing differentiation supplements, 5% FBS and 1% penicillin/streptomycin solution (MADM, Mesenchymal Stem Cell Adipogenic Differentiation Medium, ScienCell Research Laboratories, Carlsbad, CA, USA). The adipogenenic differentiation medium was replaced every 3–4 days and the extent of adipogenic differentiation was noted by microscopic observation of lipid vacuoles in the induced cells. After about 14 days, ADMSCs became mature adipocytes that could be used for transfection with siRNA.

### 2.3. Evaluation of the Cell Phenotype—Flow Cytometric Analysis

Immunophenotypic characterisation of the visceral and subcutaneous ADMSCs in passage 3 was performed using flow cytometry. ADMSCs were stained with selected antihuman fluorochrome-conjugated monoclonal antibodies: anti-CD73 FITC (clone AD2), anti-CD90 (clone SE10), anti-CD105 APC (clone 43A3), anti-CD45, anti-CD133, anti-CD10 and anti-CD141 (Biolegend). Following incubation, unbound antibodies were washed during centrifugation in PBS. Subsequently, to exclude dead cells from analysis, 7-amino-actinomycin D (7AAD) viability dye (BD Bioscience) was added for 15 min prior to acquisition. Flow cytometric data were acquired immediately after staining, with the use of a FACS Calibur flow cytometer (BD Bioscience). Proper gating strategy was established using unstained cells and FMO controls. Initially, the population of ADMSCs was distinguished on the basis of relative size (forward scatter, FSC) and relative internal structure (side scatter, SSC); media-related debris was excluded from analysis. Lack of 7AAD-related fluorescence was used to gate viable cells, and furthermore, frequency and MFI (mean fluorescence intensity) of the above-mentioned surface markers within cells of interest were evaluated (Appendix A). The percentage of positive cells are shown for each peak for markers CD105, CD73 and CD90 and for the cells lacking the negative-containing hematopoietic markers (e.g., CD45) (Appendix A). Raw flow cytometric data were processed with the use of FlowJo software (Tree Star Inc., Ashland, OR, USA) and statistically analysed in GraphPad Prism software.

### 2.4. Multilineage Differentiation Potential of ADMSCs

The functional identification of mesenchymal stem cells was done using a human mesenchymal stem cell functional identification kit (R&D Systems, Inc., Minneapolis, MN, USA) according to the manufacturer’s instructions.

#### 2.4.1. Adipogenic Differentiation Assay

For an adipogenic differentiation protocol, please see Section 2.2. The intracellular accumulation of lipids in the isolated ADMSCs was monitored by microscopic examination. In order to define the mature phenotype of the adipocytes, the cells were subjected to staining with Oil Red O and immunocytochemistry using an anti-mFABP4 antibody (see below).

#### 2.4.2. Osteogenic Differentiation Assay

ADMSCs were seeded at a density of 4.2 × 10^3^/cm^2^ into a 24-well plate in α MEM basal medium. At 50–70% confluency, the media was changed to an osteogenic differentiation medium to induce osteogenesis in the experimental wells and to the basal medium in control wells. The medium was replaced every 3–4 days until day 21. Then osteocytes were fixed with 4% paraformaldehyde for 30 min at room temperature and saved for immunostaining with anti-hOsteocalcin antibody.

#### 2.4.3. Chondrogenic Differentiation Assay

A quantity of cells, numbering 2.5 × 10^5^, were transferred to a 15 mL conical tube, centrifuged at 200× *g* for 5 min at room temperature and resuspended with 1 mL of DMEM/F-12 basal media. Then the cells were again centrifuged and resuspended in 0.5 mL of chondrogenic differentiation media. After centrifugation the caps of the tubes were loosened, and the cells were incubated upright at 37 °C and 5% CO_2_. The medium was replaced every 2–3 days, and, after 14–21 days, the chondrocyte pellet was fixed and prepared for frozen sectioning. The immunochemistry of chondrocytes, incubating the sections with the anti-hAggrecan antibody, was prepared according to the manufacturer procedures.

### 2.5. Transfection of siRNA

BLOCK-iT Fluorescent Oligo was used to monitor siRNA transfection in human adipocytes differentiated from ADMSCs and optimise the transfection protocol. We used a mixture of two distinct siRNAs for *TBC1D1* and *TBC1D4*, which refers to Silencer Select Pre-designed siRNAs (for *TBC1D1*—ID: s23308 and ID: s23307; *TBC1D4*—ID: s19140 and ID: s19142). Negative siRNA refers to Silencer Select Negative Control #1 siRNA. Human adipose-derived mesenchymal stem cells were seeded in complete medium on 12-well plates (Nunc) and then differentiated into adipocytes. The standard complexation protocol was employed, using Viromer green (Lypocalyx) according to the manufacturer’s instruction. In order to obtain the highest transfection efficiency with minimal effects on cell viability, we optimised the transfection conditions. In total, 5 groups were designed in triplicate: (i) non-transfected adipocytes differentiated from ADMSCs (untreated control, designation of the group: Ctrl); (ii) negative control siRNA (non-targeting, designation of the group: NC); (iii) adipocytes differentiated from ADMSCs transfected with *TBC1D1* siRNAs (knockdown of *TBC1D1*, designation of the group: D1KD); (iv) adipocytes differentiated from ADMSCs transfected with *TBC1D4* siRNAs (knockdown of *TBC1D4*, designation of the group: D4KD); (v) adipocytes differentiated from ADMSCs simultaneously transfected with *TBC1D4* and *TBC1D1* siRNAs (combined knockdown of *TBC1D1* and *TBC1D4*, designation of the group: D1/D4KD). The highest transfection efficacy (80–85%) was obtained using the siRNA at a concentration of 25 nM. Therefore, we used this concentration of siRNAs for the proper experiments.

### 2.6. Immunofluorescence Staining and Confocal Analysis

Cells subjected to analysis with a confocal microscope were cultured with dedicated cell-culture imaging plates (Eppendorf). Following incubation, cell-culture media were removed from the plates, and cells were fixed using CellFIX (BD Bioscience). Preliminary steps, prior to staining, included: washing with 1% BSA (Sigma-Aldrich) in PBS (Corning), permeabilisation using detergent (0.1% Triton X-100 and 0.02% SDS in PBS (Sigma-Aldrich) and blocking with 10% donkey serum (Gibco) in PBS. Next, the cells were stained with anti-human primary antibodies aimed at selected markers: anti-AS160 (dilution 1:100; rabbit IgG) (Biorbyt Ltd., Cambridge, UK), anti-CD36 (dilution 1:100; rabbit IgG) (Novus Biologicals), anti-FABP4 (dilution 1:200; rabbit IgG) (Abcam), anti-TBC1D1 (dilution 1:100; mouse IgG) (Novus Biologicals), anti-FABPpm (dilution 1:100; rabbit IgG) (Abcam), anti-Osteocalcin (10 μg/mL; mouse IgG) and anti-Aggrecan (10 μg/mL; goat IgG) (R&D Systems). Subsequent to the incubation of cells with primary antibodies, the cells were washed in 1% BSA in PBS and subjected to immunostaining with secondary antibodies conjugated to fluorochromes. Depending on the primary antibody host and isotype, the following secondary antibodies were used: anti-rabbit Alexa Fluor 488 (dilution 1:1000; goat IgG), anti-mouse Alexa Fluor 488 (dilution 1:1000; goat IgG) and anti-goat Alexa Fluor 488 (dilution 1:1000; donkey IgG) (Thermo Fisher Scientific). After incubation and washing, the cells were covered with ProLong Gold with DAPI to preserve a strong fluorescent signal and prevent fading and to stain the cell nuclei with intercalating dye—DAPI (Thermo Fisher). The acquisition of immunofluorescence data was performed using a FV1200 confocal microscope (Olympus). Additional assessment of changes in the fluorescence intensity of selected markers was implemented using ImageJ software [22], and the obtained data were statistically analysed and visualised with the use of GraphPad Prism 8 software (GraphPad Prism Software Inc., San Diego, CA, USA).

### 2.7. Oil Red O Staining

The visualisation of lipid-rich vacuoles was carried out with Oil Red O staining. Mature adipocytes were washed three times with PBS and were fixed with 10% formalin for 30 min at room temperature. Subsequently, the wells were washed, left to dry completely and stained with Oil Red O for 1 h. Wells were then washed multiple times with PBS, and images were created using an inverted microscope (Olympus, magnification, ×400). After extracting Oil Red O with 100% isopropanol, the absorbance of the extracted dye was measured spectrophotometrically at 510 nm by the use of a microplate reader (Synergy H1 Hybrid Reader, BioTek). For control, undifferentiated ADMSCs were stained with Oil Red O. Oil Red O concentration was determined against known standards of Oil Red O (0.02 mg/mL to 2.5 mg/mL).

### 2.8. Quantitative Real-Time PCR

Total RNA was isolated from the cells using the TRIzol Reagent according to the manufacturer’s instructions (Sigma Aldrich, Saint Louis, MO, USA). The quality of extracted RNA was assessed by spectrophotometry (at an absorbance OD ratio of 260/280 and 260/230) and verified by running the agarose electrophoresis with ethidium bromide. RNA that had absorbance OD ratios greater than 2 was used for downstream applications. First-strand cDNA was synthesised from 1 µg RNA using the EvoScript universal cDNA master kit (Roche Molecular Systems, Boston, MA, USA). Quantitative real time polymerase chain reaction (qRT-PCR) was carried out using the LightCycler 96 System Real-Time thermal cycler with FastStart essential DNA green master (Roche Molecular Systems) as the detection dye. Cycling conditions were: 15 s denaturation at 94 °C, 15 s annealing at 57 °C for *RPLO13A* and *TBC1D1*, 58 °C for *TBC1D4* and *CD36/SR-B2*, 59 °C for *FATP1* and *FATP4*, and 62 °C for *FABP4* and *FABPpm*, and then a 15 s extension at 72 °C for 45 cycles. Results were normalised against the housekeeper gene (*RPLO13A*) and calculated according to the Pfaffl method [23]. All samples were assayed in duplicate. The primers used in the study are listed in Table 1.

### 2.9. Plasma Membrane and Total Proteins Immunoblotting Analysis

Water containing insoluble plasma-membrane proteins fraction (PM) was isolated from adipocytes using a commercial kit according to the manufacturer’s protocol. Typical protein yields were about 30–60 µg per sample for the cell plasma-membrane fraction. The protein concentration was determined using the bicinchoninic acid (BCA) method and then the samples were collected for Western blot analysis. To confirm the purity of the plasma-membrane fraction, we assessed the expression of several specific markers. The presence of an α subunit Na^+^/K^+^ pump was confirmed exclusively in the plasma-membrane fraction. The absence of sarcoendoplasmic reticulum calcium transport ATPase (SERCA), cytochrome *c*, and glyceraldehyde 3-phosphate dehydrogenase (GAPDH) excluded contamination by other organelles (e.g., sarcoplasmic reticulum and mitochondria) or cytoplasm proteins (Figure 1).

Routine Western blotting procedures were used to detect protein content in plasma membranes and total lysate [24]. In brief, the ADMSCs were lysed in ice-cold RIPA (radioimmunoprecipitation assay) buffer containing a cocktail of protease and phosphatase inhibitors (Roche Diagnostics GmbH, Mannheim, Germany). The total protein concentration was determined using the BCA method with bovine serum albumin (BSA) as a standard. Subsequently, lysates were reconstituted in Laemmli buffer. The same amounts of protein (30 µg) were loaded on Criterion TGX Stain-Free Precast Gels (Bio-Rad, Hercules, CA, USA). After electrophoresis, proteins were transferred onto PVDF (polyvinylidene difluoride) membranes. Then, all the membranes were blocked and incubated overnight at 4 °C with the corresponding primary antibodies in appropriate dilutions. The primary antibodies were purchased from Abcam (FABP4, FATP4, FABPpm), Merck Millipore (TBC1D4), Novus Biologicals (TBC1D1) and Santa Cruz Biotechnology (GAPDH, CD36/SR-B2, FATP1, Na^+^/K^+^ pump, SERCA, Cytochrome C). Thereafter, anti-rabbit and anti-goat IgG horseradish peroxidase-conjugate secondary antibodies (1:3000; Santa Cruz Biotechnology, Dallas, TX, USA) were used. The protein bands were quantified densitometrically using a ChemiDoc visualisation system (Bio-Rad, Hercules, CA, USA). Eventually, the protein expression (optical density arbitrary units) was normalised to GAPDH or Na^+^/K^+^ pump expressions and was related to the NC group.

### 2.10. 9,10-[3H]-Palmitic Acid Uptake

Palmitic acid uptake was evaluated according to the Chavez and Summers procedure [25]. Adipocytes differentiated from ADMSCs were serum-starved for 3 h before adding Krebs–Ringer-HEPES buffer. To start the palmitate uptake, the cells were incubated with Krebs–Ringer-HEPES buffer supplemented with palmitic acid (Sigma–Aldrich) bound to fatty acid-free bovine serum albumin (Sigma–Aldrich, St. Louis, MO, USA) with the addition of radiolabelled [9,10-3H] palmitic acid (Perkin Elmer, Shelton, CT, USA) at the specific activity of 1 μCi mL for 5 min at 37 °C/5% CO_2_. Afterwards, adipocytes were placed on ice, and the reaction was terminated by adding ice-cold PBS buffer. Subsequently, the cells were washed three times with ice-cold washing buffer and solubilised in 0.05 N NaOH. Radioactivity was measured using a Packard TRI-CARB 1900 TR scintillation counter and normalised with respect to protein concentrations.

### 2.11. GLC

The contents of triacylglycerols (TAG), diacylglycerols (DAG) and free fatty acids (FFA) were quantified as described elsewhere [7]. Lipids were extracted in a chloroform–methanol solution according to the Folch method [26] and separated into different fractions using thin-layer chromatography (TLC). Identification and quantification of individual fatty acid methyl esters (FAMEs) were based on the retention times of standards using gas–liquid chromatography (GLC; Hawlett-Packard 5890 Series II gas chromatograph, HP-INNOWax capillary column). The total amount of TAG, DAG, and FFA was estimated as the sum of the particular fatty acid species in each assessed fraction. Lipid content was expressed in nanomoles per milligram of protein.

### 2.12. Statistical Analysis

The numbers of donors are mentioned in the Tables’ and Figure’ legends. Unless stated otherwise, each measurement for a single donor was performed in technical triplicate. The means of the replicates were used for further calculations. Data preprocessing was done in MS Excel, whereas statistical analyses were carried out using R, i.e., a language and environment for statistical computing (https://www.r-project.org/, accessed on 30 March 2021). Data distribution was tested with the Shapiro–Wilk test (test for normality). The homogeneity of variance was assessed with Fligner–Killeen test. The data that fulfilled normality and homogenity assumptions were analysed using the Student’s t-test, otherwise a Wilcoxon rank sum test was performed.

In order to avoid any unspecific effects of our technique (siRNA silencing with Viromer green reagent) we have employed a negative control (NC) group (containing a non-targeting siRNA fragment). Therefore, the results for the single- (*TBC1D1* or *TBC1D4*) and double-knockdowns (*TBC1D1* and *TBC1D4*) were compared with the above-mentioned group (designation of this group: NC).

## 3. Results

### 3.1. Assessment of ADMSCs Cells’ Phenotypes

In order to identify ADMSCs as adipocyte progenitors and to rule out their potential contamination by other adipose-tissue cell types, immunophenotypic characterisation was performed. To this end, the expression of specific cell surface markers was evaluated using flow cytometry. The analysis showed that our ADMSCs expressed high levels of CD90, CD105 and CD73 (the percentage of positive cells exceeded 99.6) but lacked the expression of CD45 and lineage markers (0.30% and 0.23% for CD45 and lineage markers, respectively) (Appendix A). Additionally, we checked specific depot-dependent markers of ADMSCs obtained from subcutaneous and visceral fat depots (Figure 2). Based on the mean fluorescence intensity, CD10 and CD105 were found to be the most reliable markers that allow the distinguishing of subADMSCs from visADMSCs (Figure 3). Dominant CD10 expression in subADMSCs and low in visADMSCs was consistent with studies carried out by Ong et al. [27]. Unlike CD10, CD105 was expressed at higher levels within visADMSCs. Interestingly, despite only slight differences between subADMSCs and visAMSCs, stem cell marker CD133 expression was in accordance with CD105 level between these two populations of ADMSCs. Moreover, our study indicates that ADMSCs from both depots express similar CD141 expression (Figure 3). Thus, it appears that only CD10 and CD105 can be used as prospective markers.

### 3.2. Multilineage Potential of ADMSCs Cells

To confirm the multipotent mesenchymal stem cell properties of the ADMSCs, a comparative in vitro trilineage differentiation assay was carried out. The ADMSCs from both adipose tissue depots underwent differentiation to adipocytes (as confirmed by immunocytochemistry with the anti-FABP4 antibody and Oil Red O staining), osteocytes (as corroborated by immunostaining with the anti-hOsteocalcin antibody) and chondrocytes (as validated by immunochemistry with the anti-hAggrecan antibody). The process of multilineage differentiation of ADMSCs was weekly monitored for three subsequent weeks (Appendix A). After osteogenic differentiation, the cells showed fusiform morphology. Their mineralisation was visualised using immunostaining with the anti-hOsteocalcin antibody. The formation of round or oval nodules has been observed after chondrocyte differentiation. After differentiation, the cells were characterised by the presence of FABP4, a marker specific to human adipocytes, located in the cytoplasmic compartment where lipid droplets are found (Appendix A). Moreover, Oil Red O staining demonstrated that the adipogenically stimulated cells displayed numerous lipid-filled vacuoles, whereas no lipid accumulation was found in control, undifferentiated cells. Spectrophotometric quantification revealed a few fold increase in the concentration of intracellular lipids for both subcutaneous (+1.56-fold, *p* < 0.05, for the differentiated versus the undifferentiated group) and visceral cells (+2.98-fold, *p* < 0.05, for the differentiated versus the undifferentiated group) (Figure 4). As a point of notice, the adipocytes differentiated from visADMSCs exhibited substantially lower FABP4 protein content than the adipocytes from subADMSCs (−54%, *p* < 0.05, Appendix A).

### 3.3. Efficiency of TBC1D1 and TBC1D4 Gene Silencing in the Adipocytes Differentiated from ADMSCs

To study the biological function of the two signaling proteins TBC1D4 and TBC1D1, the adipocytes differentiated in vitro from ADMSCs were transfected with siRNA against *TBC1D4* or *TBC1D1*. More precisely, the cells were transfected with BLOCK-iT Fluorescent Oligo siRNA using Viromer green as a carrier. Fluorescence was predominantly noted in the nuclei of the cells (data not shown). To validate the successful knockdown of the two target genes, the expression of *TBC1D1* and *TBC1D4* mRNA and protein was assessed via RT-PCR and Western blot analysis. In all the analysed transfected adipocytes, the abundance of TBC1D4 was not altered in TBC1D1-deficient cells. Similarly, TBC1D1 expression was not changed in the adipocytes deficient in TBC1D4 (Figure 5).

The knockdown efficiency in *TBC1D1* and *TBC1D4* mRNA was 79% and 81% in adipocytes from subADMSCs and 60% and 84% in adipocytes from visADMSCs (*p* < 0.05, Figure 5A,B). Additionally, the protein expression of TBC1D1 and TBC1D4 was diminished in the transfected cells (*p* < 0.05, Figure 5C,D). The fluorescence results confirm that the protein levels of TBC1D1 and TBC1D4 were decreased in siRNA-mediated knocked groups in the adipocytes differentiated from ADMSCs (Appendix A).

### 3.4. Fatty Acids Transport Proteins Expression after TBC1D4 and TBC1D4 Silencing in the Adipocytes Differentiated from ADMSCs

To determine whether TBC1D1 and TBC1D4 deficiency influenced fatty acid transport, we measured the plasmalemmal and total expression of fatty acids handling proteins in the adipocytes differentiated from the subcutaneous (subADMSCs) and visceral (visADMSCs) adipose tissue depots.

#### 3.4.1. Fatty Acid Transporter Changes in Adipocytes Differentiated from subADMSCs

Neither mRNA nor the total protein expression of CD36/SR-B2 were significantly altered in any of the transfected groups in comparison to the control (Figure 6B,C). The subcellular protein distribution of CD36/SR-B2 was changed in the case of the D4KD and D1/D4KD group but not the D1KD group (Figure 6A). Namely, plasma-membrane CD36/SR-B2 localisation was increased upon TBC1D4 deficiency, compared with the control cells (+64% and +72%, for D4KD and D1/D4KD versus NC, *p* < 0.05, Figure 6A).

Total *FABPpm* mRNA levels were decreased only in the TBC1D1-deficient adipocytes (−41%, *p* < 0.05, Figure 7C). Plasmalemmal FABPpm protein expression was significantly higher only in the D1/D4KD group (+59%, *p* < 0.05, Figure 7A). Comparing the two single knocked-down groups, we noticed an increased expression of FABPpm at the plasma-membrane level for the D4KD group (+33% for D4KD versus D1KD, *p* < 0.05, Figure 7A).

As illustrated in Figure 8C, the knockdown of Rab-GTPase-activating proteins did not significantly alter mRNA content for *FATP1*. It did, however, decrease the total content of FAPT1 protein, a result that reached the level of statistical significance in the case of D1/D4KD (*p* < 0.05, Figure 8B). Plasmalemmal FATP1 protein expression was not affected by *TBC1D1* and/or *TBC1D4* knockdown in the adipocytes differentiated from subADMSCs (Figure 8A).

TBC1D4- and/or TBC1D1-deficient adipocytes showed FATP4 mRNA levels comparable to the level found in the negative control (Figure 9C). Also total FATP4 protein expression remained virtually unchanged between the subADMSC-derived adipocytes (Figure 9B). Plasmalemmal content of FATP4 was elevated only in the case of the D4KD group (+36%, D4KD vs. NC, *p* < 0.05, Figure 9A). Additionally, we noticed some changes between the knocked-down groups. Double-deficient adipocytes showed an 18% reduction in their plasmalemmal content (D1/D4KD vs. D4KD, *p* < 0.05, Figure 9A).

#### 3.4.2. Fatty Acid Transporter Changes in Adipocytes Differentiated from visADMSCs

Although no significant differences were seen in the mRNA expression of *CD36/SR-B2*, we detected elevated total protein content in the D4KD and D1/D4KD group in the adipocytes differentiated from visADMSCs (+137% and +122%, for D4KD and D1/D4KD versus NC, respectively, *p* < 0.05, Figure 6B). Moreover, TBC1D4 deficiency substantially elevated the plasmalemmal content of CD36/SR-B2 (+116%, D4KD versus NC, *p* < 0.05, Figure 6A).

As revealed by real-time PCR (Figure 7C) and Western blot (Figure 7B) methods, the knockdown of *TBC1D1* and/or *TBC1D4* did not significantly change total FABPpm expression at the transcript or protein level in the adipocytes. Regarding the plasmalemmal content of FABPpm, we noticed that it was elevated in TBC1D4 deficient adipocytes (+34%, *p* < 0.05, Figure 7A). Furthermore, we found that the double-knocked-down adipocytes exhibited lower plasma-membrane FABPpm content in comparison with the negative control and single siRNA transfected cells (*p* < 0.05, Figure 7A).

The total protein expression of FATP1 was diminished in *TBC1D1*-transfected cells (*p* < 0.05, Figure 8B). There were no significant differences between the groups as regards to *FATP1* at mRNA, as well as at plasma-membrane levels, in the adipocytes derived from visADMSCs (Figure 8A,C).

Transfected adipocytes exhibited an unaltered *FATP4* mRNA level (Figure 9C). However, *TBC1D1* knockdown, as well as the simultaneous knockdown of both the Rab-GTPase-activating proteins, caused a decrease in the total FATP4 protein content (*p* < 0.05, Figure 9B).

#### 3.4.3. Tissue-Specific Differences between the Adipocytes Derived from Subcutaneous and Visceral Adipose Tissue

Importantly, we noticed some differences in the fatty acid transporters localisation/expression between the adipocytes of subcutaneous and visceral origin (visNC versus subNC). Namely, the total protein and plasmalemmal content of CD36/SR-B2 were markedly greater in the adipocytes differentiated from subADMSCs (+5.51-fold and +2.1-fold, for total subNC versus total visNC, and plasmalemmal subNC versus plasmalemmal visNC, respectively, *p* < 0.05, Figure 6A,B). These results were similar to the abundancy and subcellular localisation of the FA transporters analysed by immunofluorescence staining (Appendix A). Fatty acid transporters in the adipocytes were dispersed throughout the cell interior and plasma membrane. Moreover, the total protein expression of FATP1 was significantly greater in the control adipocytes derived from subcutaneous adipose tissue (+3.19-fold for total subNC versus total visNC, Figure 8B). The adipocytes of visADMSC origin exhibited higher *FABPpm* content at transcript level (+1.05-fold for mRNA visNC versus mRNA subNC, *p* < 0.05, Figure 7C).

### 3.5. Palmitate Uptake in TBC1D4- and TBC1D1-Deficient Adipocytes Differentiated from ADMSCs

Our data demonstrate that both the single- (D4KD) and double-knockout group (D1/D4KD) had significantly increased basal palmitate uptake in the adipocytes derived from subADMSCs, as compared to the control group (+66% and +50%, for D4KD and D1/D4KD versus NC, *p* < 0.05, Figure 10). Given the above, we observed markedly elevated basal palmitate uptake in the groups with knocked-down TBC1D4 (+77% and +1.01-fold, for D4KD and D1/D4 versus D1KD, respectively, *p* < 0.05, Figure 10).

### 3.6. The Content and Composition of Selected Lipid Fractions in the Adipocytes Differentiated from ADMSCs with TBC1D4 and/or TBC1D1 Deficiency

The total content of free fatty acids (FFA), diacylglycerols (DAG) and triacylglycerols (TAG) remained relatively unchanged between all of the experimental groups (Figure 11). 

## 4. Discussion

At the outset of our experiment, we assessed the appropriateness of our model. First, we evaluated the properties of the ADMSCs obtained from the subcutaneous and visceral adipose tissue of lean patients. We sought to determine whether the collected cells fulfilled the criteria for mesenchymal stem cells proposed by the International Society for Cell and Gene Therapy (ISCT) [28]. In accordance with the guidelines, we noticed that our cells: (1) were plastic adherent; (2) expressed a set of required antigens (CD105, CD90, CD73) and were devoid of CD45, CD133, as well as hematopoietic and endothelial lineage markers; (3) were capable of differentiating in vitro into osteoblasts, chondrocytes and adipocytes (trilineage potential). Overall, the obtained data confirm the appropriateness of our experimental model, that is, yielding the desired ADMSCs.

All the adipocytes in our experiment were obtained from female donors. This fact is of high importance, because there are sex-based differences in body-fat distribution and adipose-tissue metabolism between lean men and women. In fact, Shadid et al. reported that direct FFA uptake was about 70% greater in subcutaneous adipose tissue of normal-weight women than in the same location in men [29]. Accordingly, Edens et al. found that the rate of TAG synthesis is higher in women in subcutaneous adipose tissue compared to their abdominal depot or any location in men [30]. Thus, in the current study, we examined only the cells obtained from postmenopausal women. Previous investigations demonstrated that white adipose tissue can be divided into a few depots characterised by distinctive physiological properties. Metabolic differences between subcutaneous and visceral adipose tissue are stable and persist after ADMSC isolation and also in the adipocytes’ cell culture. For instance, we observed a greater palmitate uptake in the cells of subcutaneous origin (+49%). The above was most likely caused by greater total and plasma-membrane expression of some fatty acid transport proteins, mainly: CD36/SR-B2 (+2.1-fold) and FATP1 (+1.49-fold). Additionally, non-transfected adipocytes differentiated from subADMSCs had higher FABP4 expression compared to visADMSCs, which could contribute to greater lipid accumulation in the cytoplasm.

Two closely related Rab-GTPase-activating proteins, TBC1D1 and TBC1D4, play an important, although less investigated, role in lipid handling. It has been shown that knockout mice devoid of either TBC1D1 or TBC1D4, as well as double-deficient (D1/4KO) mice, had switched to lipid usage as evidenced by the increased whole-body fat oxidation rate and decreased carbohydrate consumption [9]. Recently, both RabGAPs have emerged as proteins regulating fatty acids intracellular transport. The above is based on a few scientific observations. First, TBC1D1 and TBC1D4 are expressed in insulin-responsive tissues that are important storage sites for glucose and fatty acids. Second, the two regulators contain TBC domain, a protein motif characteristic for molecules engaged in vesicle trafficking. Although AS160 and TBC1D1 polypeptide chains share only 60% similarity, their RabGAP domains are 91% alike and display the same Rab substrate specificity in vitro, which suggests that the two proteins may have some overlapping functions [16,17].

In our study, we found that the plasmalemmal expression of fatty acid transporter CD36/SR-B2 was elevated in the TBC1D4-deficient adipocytes differentiated from both vis- and subADMSCs, while in visADMSCs, higher FABPpm build-up into plasmalemma was noticed. Additionally, in the adipocytes of subADMSC origin, the double knockdown of AS160 and TBC1D1 led to an increase in the plasmalemmal CD36/SR-B2 and FABPpm content. Importantly, their total expressions remained relatively stable. Only the total amount of CD36/SR-B2 in the adipocytes from visceral tissue was higher. Thus, plasmalemmal CD36/SR-B2 content may be increased with (adipocytes from visADMSCs) or without (adipocytes from subADMSCs) concomitant changes in its total protein expression. The latter observation clearly points to intracellular translocation of the fatty acid transporter. These results underline the importance of both proteins (CD36/SR-B2 and FABPpm) in the regulation of fatty acid transport in AS160-deficient adipocytes from the ADMSCs of lean patients. Of the two, CD36/SR-B2 was more pivotal to the process, given the more apparent expression pattern and greater rate of changes in response to *TBC1D4* knockdown (about three times greater). Our observations are in line with the previous results obtained on Zucker rats by Luiken and co-workers [31]. In that experiment, the animals had 1.8-fold elevated fatty acid uptake in adipose tissue [31]. Importantly, the increased fatty acid influx was attributed to an increase in CD36/SR-B2 and FABPpm expression, especially at mRNA and plasma-membrane levels, with no (FABPpm) or moderate changes in the total transporter levels [31,32]. This pattern resembles our present data, with the exception of mRNA, where we found virtually no statistically significant changes in the transporters transcript levels after the RabGAPs silencing. Since *CD36/SR-B2* mRNAs remained unchanged, the observed alteration in its total protein content was most likely caused by some post-transcriptional mechanism(s). However, the mechanism by which RabGAPs can alter expression levels of the transporters is not fully understood and needs to be investigated. To provide experimental support for the observed fatty acid transporters redistribution, we assessed the cellular fatty acid influx. We noted an increased basal uptake of palmitate, i. e. the most common long chain fatty acid (C16:0) found in animals, upon AS160 suppression (single- and double-knocked-down group). This effect occurred in subADMSCs but not in the adipocytes of visceral origin. The reasons behind this disparity might lay in the specific intrinsic abilities of cells derived from different fat storage sites to respond to metabolic challenges [33]. Specifically, subcutaneous fat has a greater capacity to uptake fatty acids and serves as a short-term deposit for triacylglycerols [34]. Therefore, an increase in palmitic acid uptake in our study was more readily evident in subcutaneous adipocytes. The adaptive upregulation of free fatty acids clearance by subcutaneous fat might contribute to improved insulin sensitivity [33]; apparently, the cells’ characteristic features were also sustained in in vitro conditions. In contrast, lipolytic activity predominates within visceral adipocytes, but the release of free fatty acids strengthens proportionally to body fat mass [35]. It confirms that the metabolic phenotype of subcutaneous and visceral adipocytes is preserved in in vitro conditions.

As regards other fatty acid transporters, their levels were stable no matter the experimental intervention. Apart from the visceral D1KD and D1/D4KD groups, no significant changes in the total expression of FATP4 were detected when the groups were compared with control. This is consistent with the results from murine adipocytes, where FATP4 did not show increased expression in the plasma membrane in response to physiological stimuli (e.g., insulin) [36]. On the other hand, Benninghoff et al. have shown that SLC27A4/FATP4, but not CD36/SR-B2, specifically regulates Rab-GAP-dependent FA uptake into skeletal muscle [17]. Furthermore, we have shown that *TBC1D1*-siRNA-transfected adipocytes exhibited unaltered plasmalemmal, as well as total, CD36/SR-B2 and FABPpm expression. In line with that notion, TBC1D1-deficient rats also showed unchanged total amount and subcellular localisation of CD36/SR-B2 in skeletal muscle [22]. Double knockdown (D1/D4KD) did not exert additive effects on fatty acid uptake and/or the expression of FA-transporting proteins in comparison to D4KD alone. This suggests that, in adipose tissue, TBC1D1 serves slightly different, non-overlapping roles than TBC1D4 in the regulation of the tissue lipid metabolism. In other words, neither of the Rab GAP proteins compensate for the loss of their counterpart. Accordingly, Lansey et al. [11] had previously found no change in the level of TBC1D1 in primary adipocytes isolated from AS160 knockout (AS160−/−) mice. A lack of changes in fatty acid transport after TBC1D1 downregulation may further suggest that mostly AS160 controls FA uptake under basal conditions in the differentiated adipocytes.

To sum up, our results indicate that fatty acid trafficking in adipocytes is regulated solely by TBC1D4 (AS160). We found that, in contrast to the TBC1D1-deficient adipocytes, there is a significant increase in plasmalemmal CD36/SR-B2 in TBC1D4- or double-deficient adipocytes. In addition, the changes in plasmalemmal CD36/SR-B2 content coincided with an increase in the total protein abundance in the adipocytes derived from visADMSC. Overall, our study provides evidence that, in human adipocytes, AS160 has an apparent function in fatty acid transporter dynamics.

## Figures and Tables

**Figure 1 cells-10-01515-f001:**
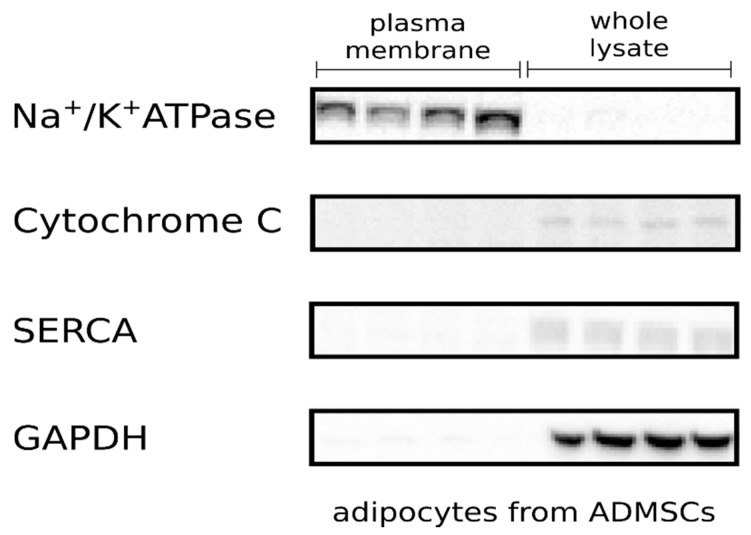
Representative Western blots showing the distribution of the Na^+^/K^+^ ATPase, SERCA, cytochrome C and GAPDH in the plasma membrane and whole lysate.

**Figure 2 cells-10-01515-f002:**
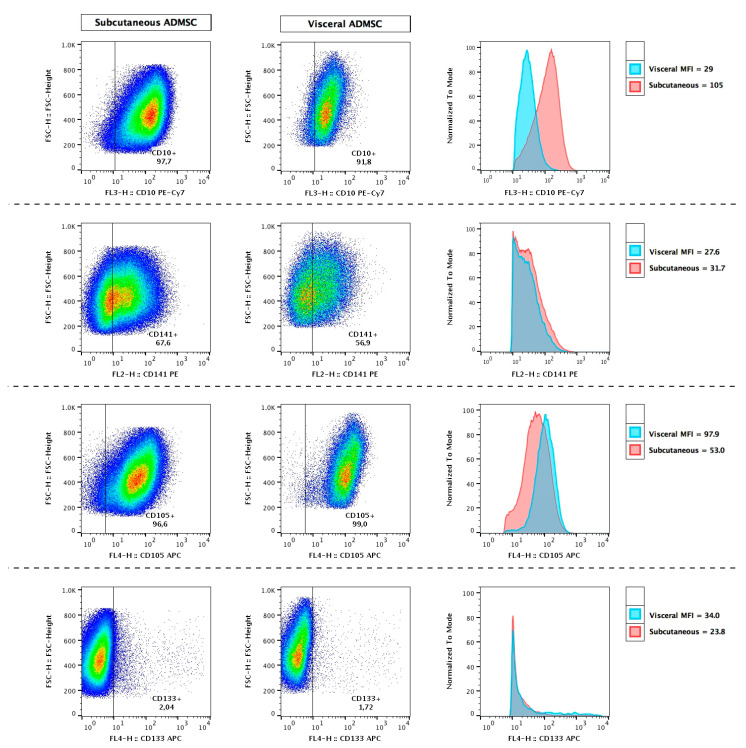
Immunophenotypic characterisation of subcutaneous and visceral ADMSCs in passage 3. Cells were stained and characterised based on CD10+, CD105+, CD133+ and CD141+ markers using flow cytometry. Numbers indicate the percentage of stained cells in the population.

**Figure 3 cells-10-01515-f003:**
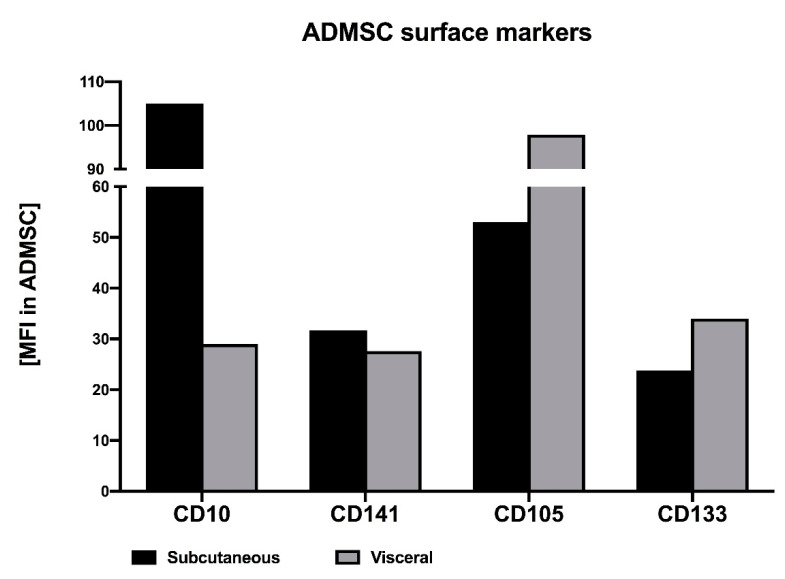
Mean fluorescent intensity (MFI) for CD10+, CD105+, CD133+ and CD141+ markers of ADMSCs obtained from subcutaneous and visceral fat depots.

**Figure 4 cells-10-01515-f004:**
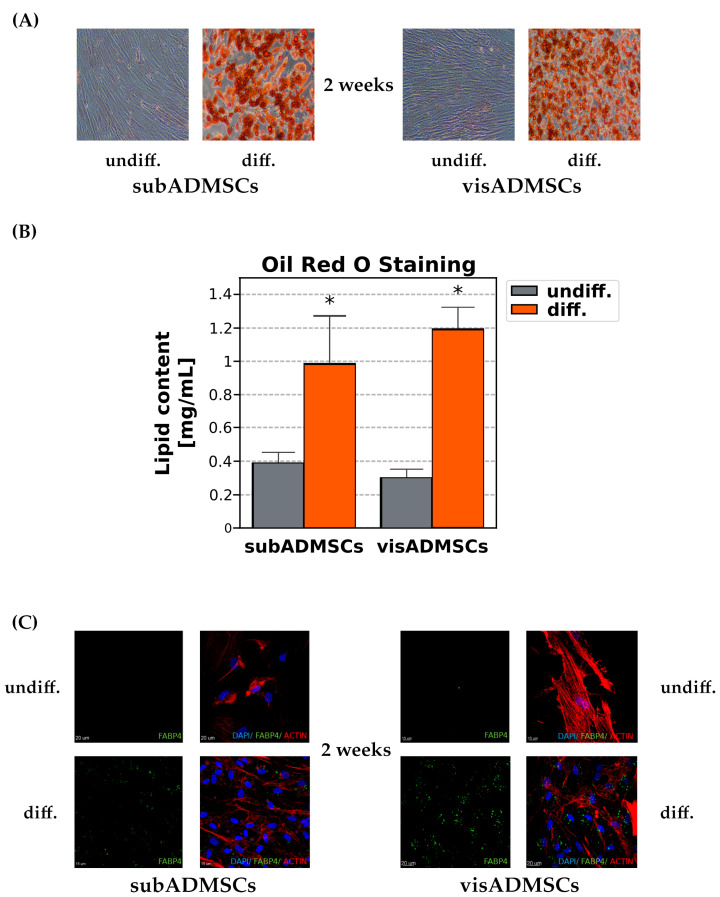
Adipocyte differentiation of subcutaneous and visceral ADMSCs. (**A**) Morphology of undifferentiated and adipogenically differentiated ADMSCs. Mature adipocytes were observed after 14 days of adipogenic differentiation from sub- and visADMSCs. Lipid vacuole presence was verified by Oil Red O staining. Magnification 10×. (**B**) Quantification of lipid content in the undifferentiated and differentiated ADMSCs. Bars and whiskers represent the mean and SD, respectively. Number of patients equals four (measurements taken in triplicate). * significantly different from undifferentiated cells, *p* < 0.05. (**C**) Intracellular localisation of FABP4 (green), nuclei (blue) and actin filaments (red) were visualised by fluorescence microscopy. Scale bar represents 15 and 20 µm.

**Figure 5 cells-10-01515-f005:**
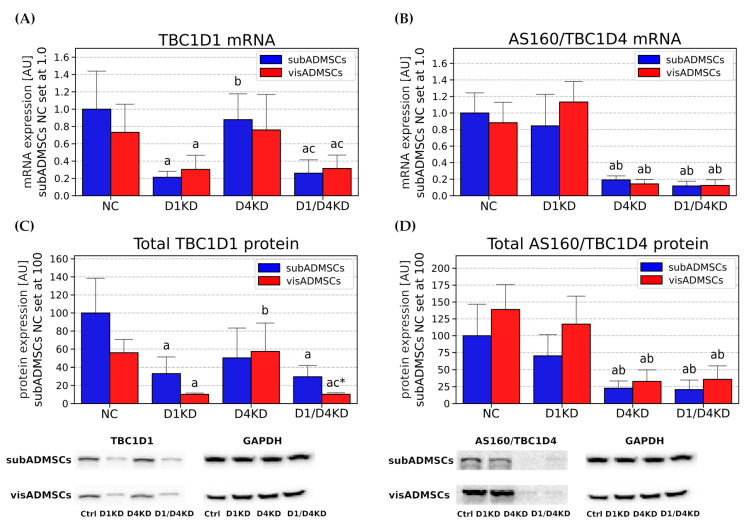
Knockdown efficiency for both Rab-GAPs (*TBC1D1* and *TBC1D4*) at mRNA (**A**,**B**) and protein level (**C**,**D**). Bars and whiskers represent the mean and SD, respectively. Number of patients equals four (measurements taken in duplicate). Values are expressed in arbitrary units; the mean in subADMSCs’ negative control was set at 100. Representative Western blots are shown. a—difference versus NC (*p* < 0.05); b—difference versus D1KD (*p* < 0.05); c—difference versus D4KD (*p* < 0.05); * *p* < 0.05 significantly different from visADMSCs. Designation of the groups: NC—negative control containing non-targeting siRNA fragment; D1KD—knockdown of *TBC1D1*; D4KD—knockdown of *TBC1D4*; D1/D4KD—double knockdown of *TBC1D1* and *TBC1D4*.

**Figure 6 cells-10-01515-f006:**
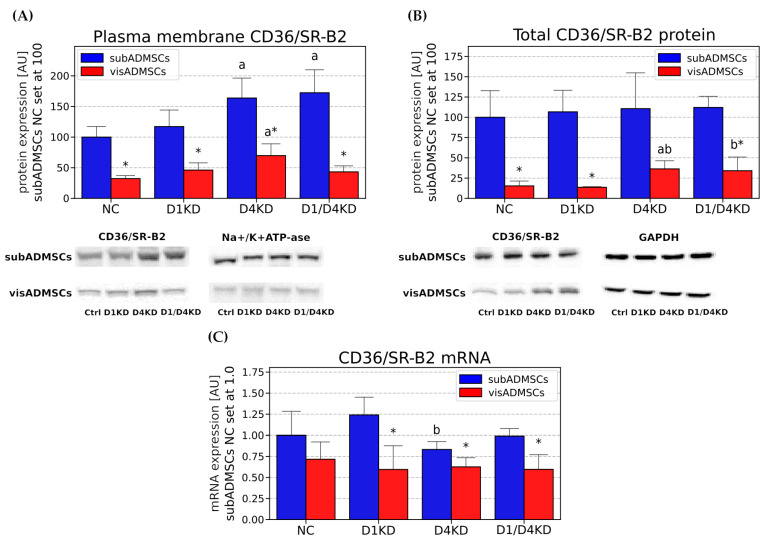
Plasma membrane (**A**), total protein (**B**) and mRNA (**C**) content of CD36/SR-B2 in TBC1D1- and/or TBC1D4-deficient adipocytes. Bars and whiskers represent the mean and SD respectively. Number of patients equals four (measurements taken in duplicate). Values are expressed in arbitrary units; the mean in subADMSCs’ negative control was set at 100. Representative Western blot images are shown. a—difference versus NC (*p* < 0.05); b—difference versus D1KD (*p* < 0.05); c—difference versus D4KD (*p* < 0.05); * *p* < 0.05 significantly different from visADMSCs. Designation of the groups: NC—negative control containing non-targeting siRNA fragment; D1KD—knockdown of *TBC1D1*; D4KD—knockdown of *TBC1D4*; D1/D4KD—double knockdown of *TBC1D1* and *TBC1D4*.

**Figure 7 cells-10-01515-f007:**
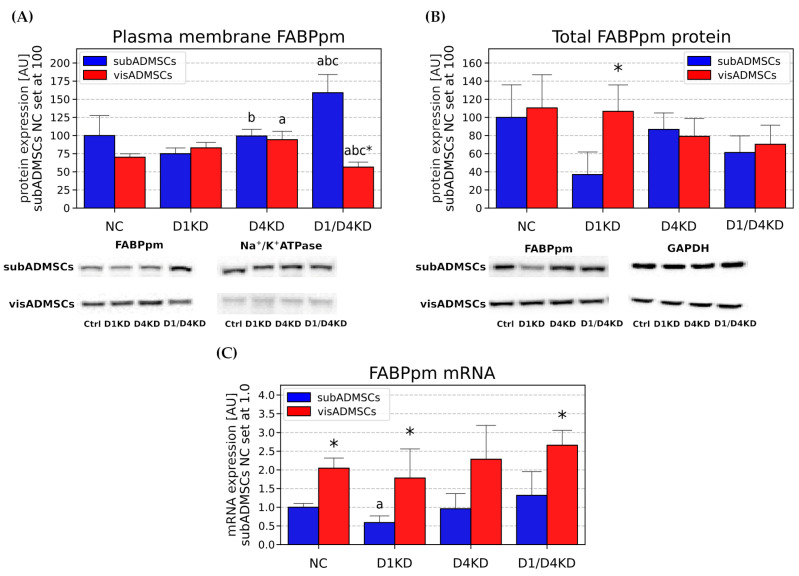
Plasma membrane (**A**), total protein (**B**) and mRNA (**C**) content of FABPpm in TBC1D1- and/or TBC1D4-deficient adipocytes. Bars and whiskers represent the mean and SD respectively. Number of patients equals four (measurements taken in duplicate). Values are expressed in arbitrary units; the mean in subADMSCs’ negative control was set at 100. Representative Western blot images are shown. a—difference versus NC (*p* < 0.05); b—difference versus D1KD (*p* < 0.05); c—difference versus D4KD (*p* < 0.05); * *p* < 0.05 significantly different from visADMSCs. Designation of the groups: NC – negative control containing non-targeting siRNA fragment; D1KD—knockdown of *TBC1D1*; D4KD—knockdown of *TBC1D4*; D1/D4KD—double knockdown of *TBC1D1* and *TBC1D4*.

**Figure 8 cells-10-01515-f008:**
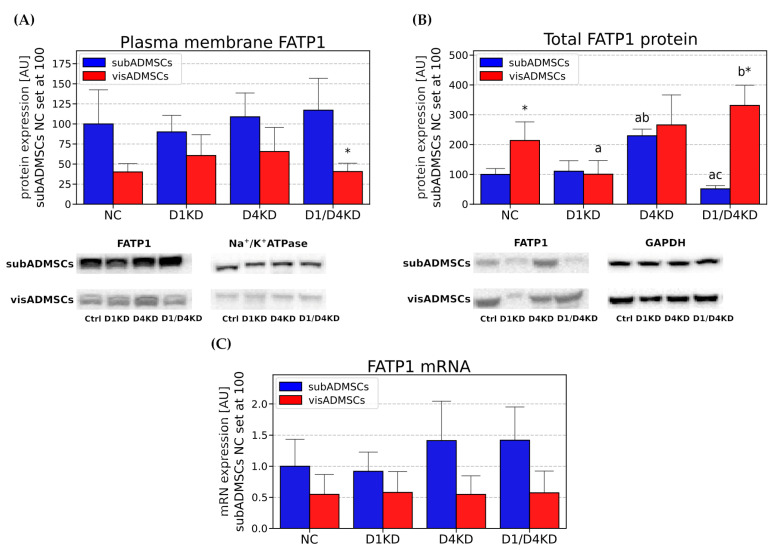
Plasma membrane (**A**), total protein (**B**) and mRNA (**C**) content of FATP1 in TBC1D1- and/or TBC1D4-deficient adipocytes. Bars and whiskers represent the mean and SD respectively. Number of patients equals four (measurements taken in duplicate). Values are expressed in arbitrary units; the mean in subADMSCs’ negative control was set at 100. Representative Western blot images are shown. a—difference versus NC (*p* < 0.05); b—difference versus D1KD (*p* < 0.05); c—difference versus D4KD (*p* < 0.05); * *p* < 0.05 significantly different from visADMSCs. Designation of the groups: NC—negative control containing non-targeting siRNA fragment; D1KD—knockdown of *TBC1D1*; D4KD—knockdown of *TBC1D4*; D1/D4KD—double knockdown of *TBC1D1* and *TBC1D4*.

**Figure 9 cells-10-01515-f009:**
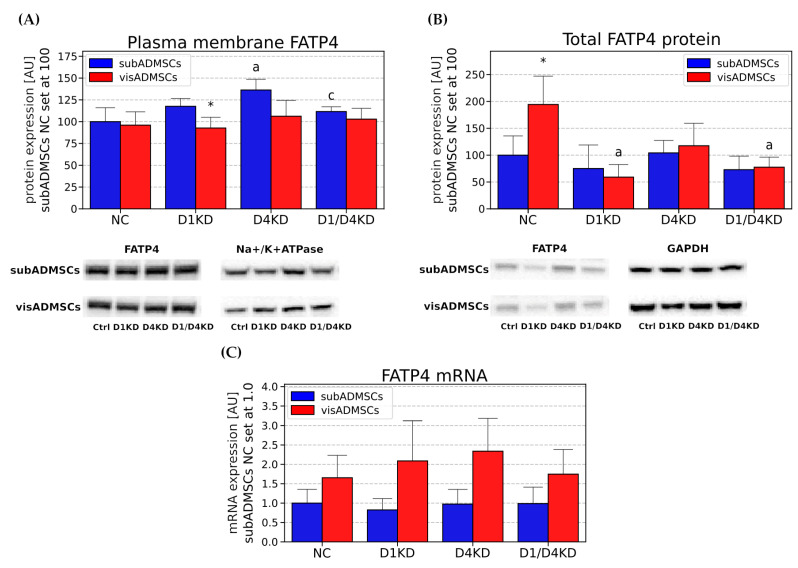
Plasma membrane (**A**), total protein (**B**) and mRNA (**C**) content of FATP4 in TBC1D1- and/or TBC1D4-deficient adipocytes. Bars and whiskers represent the mean and SD respectively. Number of patients equals four (measurements taken in duplicate). Values are expressed in arbitrary units; the mean in subADMSCs’ negative control was set at 100. Representative Western blot images are shown. a—difference versus NC (*p* < 0.05); b—difference versus D1KD (*p* < 0.05); c—difference versus D4KD (*p* < 0.05); * *p* < 0.05 significantly different from visADMSCs. Designation of the groups: NC—negative control containing non-targeting siRNA fragment; D1KD—knockdown of *TBC1D1*; D4KD—knockdown of *TBC1D4*; D1/D4KD—double knockdown of *TBC1D1* and *TBC1D4*.

**Figure 10 cells-10-01515-f010:**
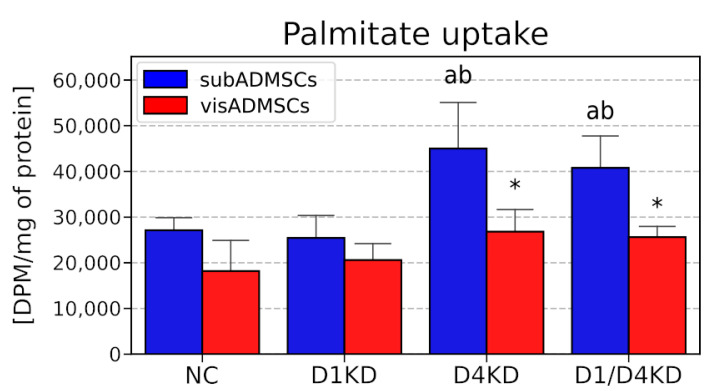
LCFA (palmitate) uptake into adipocytes after siRNA-mediated TBC1D1 and/or TBC1D4 knockdown. Bars and whiskers represent the mean and SD, respectively. Number of patients equals four (measurements taken in triplicate). Values are expressed in DPM per mg of protein. a—difference versus NC (*p* < 0.05); b—difference versus D1KD (*p* < 0.05); * *p* < 0.05 significantly different from visADMSCs. Designation of the groups: NC—negative control containing non-targeting siRNA fragment; D1KD—knockdown of *TBC1D1*; D4KD—knockdown of *TBC1D4*; D1/D4KD—double knockdown of *TBC1D1* and *TBC1D4*.

**Figure 11 cells-10-01515-f011:**
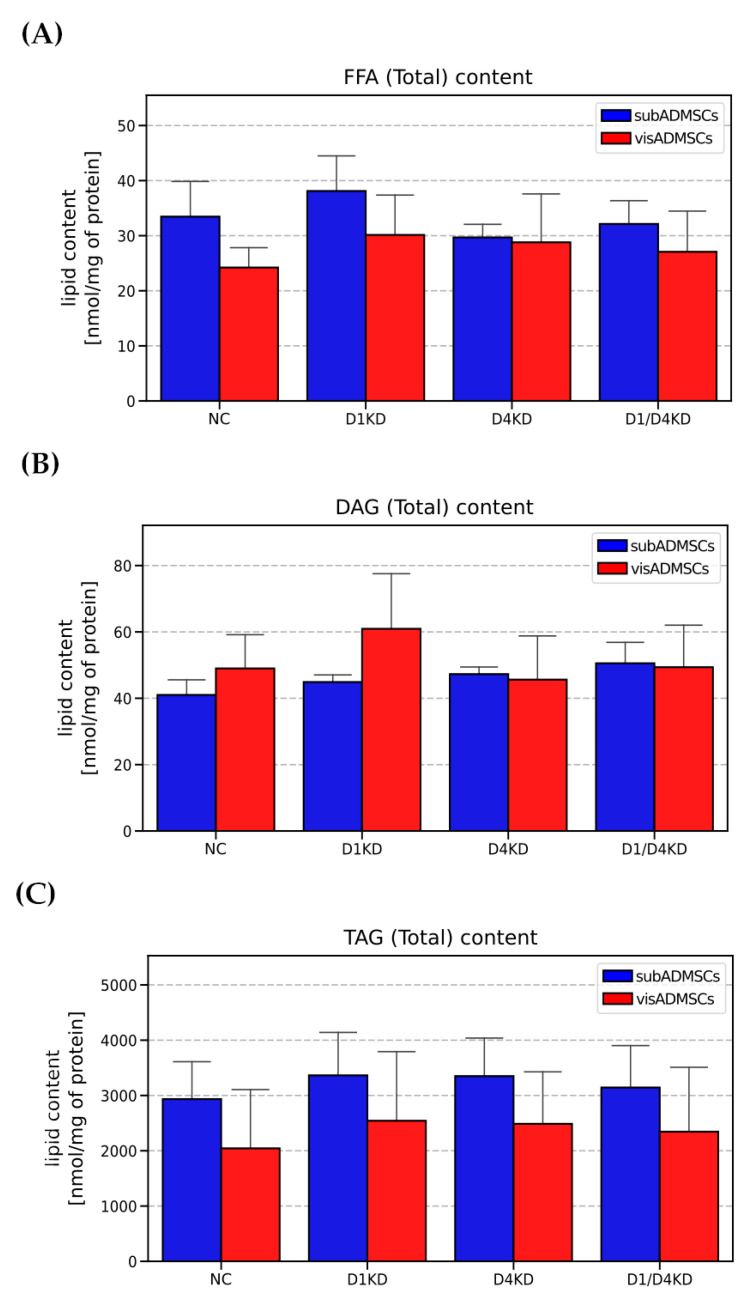
Total FFA (**A**), DAG (**B**), and TAG (**C**) content in TBC1D1- and/or TBC1D4-deficient adipocytes. Bars and whiskers represent the mean and SD respectively. Number of patients equals four (measurements taken in triplicate). Values are expressed in nmol per mg of protein. Designation of the groups: NC—negative control containing non-targeting siRNA fragment; D1KD—knockdown of *TBC1D1*; D4KD—knockdown of *TBC1D4*; D1/D4KD—double knockdown of *TBC1D1* and *TBC1D4*.

**Table 1 cells-10-01515-t001:** The primers for real-time PCR.

Target Gene	Forward Primer (5′-3′)	Reverse Primer (5′-3′)
*TBC1D4*	AGCTCCAGTGAACAGTGCAGTG	CACTTAGGGACTCATTGCTGC
*TBC1D1*	GTGTGGGAAAAGATGCTTAGCA	GTGATGACGTGGCACACCTT
*FABPpm*	GAAGGCAAAGGTGCGACAGT	GCCGAACGGTAGAGGCAAA
*FATP1*	GCTAAGGCCCTGATCTTTGG	CCAAGTCTCCAGAGCAGAAC
*FATP4*	TGGCGCTTCATCCGGGTCTT	CGAACGGTAGAGGCAAACAA
*FABP4*	GGTACCTGGAAACTTGTCTC	TTAGGTTTGGCCATGCCAGC
*CD36/SR-B2*	GGTACAGATGCAGCCTCATT	AGGCCTTGGATGGAAGAACA
*RPLO13A*	CTATGACCAATAGGAAGAGCAACC	GCAGAGTATATGACCAGGTGGAA

## Data Availability

The data presented in this study are available on request.

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
