# Peer review of "Does TBC1D4 (AS160) or TBC1D1 Deficiency Affect the Expression of Fatty Acid Handling Proteins in the Adipocytes Differentiated from Human Adipose-Derived Mesenchymal Stem Cells (ADMSCs) Obtained from Subcutaneous and Visceral Fat Depots?"

_cells, 2021, doi:10.3390/cells10061515_

Round 1
Reviewer 1 Report
In general, the authors did a lot of work, both experimentally and textually, which is appreciated. Yet, there are some unresolved issues left.
- Major comment 1: Introduction has been shortened, but is still a bit lengthy. However, this reviewer will not ask for further shortening. In one instance, information went lost. The observation that AS160-KD resulted in increased basal GLUT4 translocation to the PM [12] should be kept as information for the reader.
- Major comment 3: The resolution of the figures has increased, but the font size of the text in the figures is still small. There is still sufficient room to increase the font size (especially the X-axis). Fig. 11 is a good example of how all figures should be.
- ‘Other’ comment 5 about using GLUT4 for testing the fractionation method. This fractionation method plays a crucial role in this study. Yet, this reviewer still does not have complete confidence in the commercial fractionation method. The more extensive testing whether markers from other membrane compartments would contaminate the PM fraction (see reply to major comment 4) is appreciated, but not sufficient. Commercial companies promise a lot with respect to their products, but at the work floor these products often do not live up to the promises made. The very best way to test this procedure is to assess in a dynamic manner the effect of a (genetic) manipulation with an established action. Hence, this reviewer asks again if the authors would test the effect of AS160-KD on basal GLUT4 translocation. The knockdown should show an increase in GLUT4 content in the PM compared to the mock-transfected control. The GLUT4 Western blot added to the reply is appreciated, but not really helpful in the context of testing the fractionation method.
- ‘Other’ comment 7 about explaining role of RabGAPs in fatty acid transporter expression levels: These explanations seem far-fetched or are difficult to comprehend. Are these merely speculations from the authors or actually described in the literature? In case of speculation, the authors should simply state that this role of the RabGAPs is incompletely understood. In case of description in literature, the authors should rephrase and explain better.
Reviewer 2 Report
I'm not convinced by the answers: clearly cells are not well differentiated (only little accumulation of lipids). Thus it is impossible to conclude to an effect in adipocytes vs non-differentiated cells.
also deletion of the results concerning the "cytosolic fraction" because they can not explained what happens in it is not sientifically correct.
Reviewer 3 Report
The revised version of the manuscript by Mikłosz et al. has been significantly improved. All my questions have been adequately answered.
Author Response
The revised version of the manuscript by Mikłosz et al. has been significantly improved. All my questions have been adequately answered.
Dear Reviewer 3
Thank You very much for the second analysis of our study and Your favorable comment. We are glad that You find our paper worthy to be accepted in its present form.
Best regards, Agnieszka Miklosz – corresponding author
This manuscript is a resubmission of an earlier submission. The following is a list of the peer review reports and author responses from that submission.
Round 1
Reviewer 1 Report
This study describes the role of the RabGAPs TBC1D4 and TBC1D1 in regulation of fatty acid uptake into stem cell-derived human adipocytes. This is an important question because information about regulation of fatty acid uptake can be used to further optimize stem cell differentiation protocols. Additionally this information is relevant for possible novel therapeutic strategies to treat obese/diabetic patients. It is appreciated that no less than 4 subjects were used for generation of adipocytes from stem cells, which must have included a lot of culturing. Furthermore, the multilineage potential of the adipocyte stem cells has been extensively characterized. Despite the timeliness of the topic, there are several shortcomings in the present study.
Major Comments:
- Introduction: By far too long and not to the point. It is written in the style of a review. The writing style should be much more compact and sentences should be grammatically checked. The authors should attempt to shrink the Intro at least by half. Figure 1 is redundant and can be deleted.
- The subADMSCs and the visADMSCs should be characterized by accepted marker proteins discriminating both adipose cell types. Such characterization is absent now.
- Figs 4-8: The letter type used in the panels of all these figures is much too small. Also the pixel quality is very bad. Hence, the text in these panels is impossible to read.
- Fractionation procedure. Translocation of fatty acid transporters occurs from subcellular membrane compartments to the plasma membrane. All fatty acid transporters are membrane proteins, and are never to be found in the cytosol, i.e., in water-soluble fractions. Therefore, the selected fractionation procedure, including a cytosolic fraction, is inappropriate. A much more appropriate fractionation procedure for studying transporter translocation would allow the separation of intracellular membranes from the plasma membrane. Yet, as in the case of GLUT4 translocation, it is sufficient to show only the increase in plasma membrane content this transporter, as carried out in most publications on this topic. Hence, all data concerning the cytoplasmic fraction in Figs. 4-8 should be deleted. But still, the plasma membrane fraction should be much better characterized by marker proteins. Such characterization is absent now. This characterization should at least include enrichment of established marker plasma membrane proteins in the PM fraction compared to the lysate used for fractionation. Furthermore, it should be shown that marker proteins for intracellular membranes are absent in the PM fraction.
- Non-significant trends: Throughout the Results and discussion section, multiple non-significant changes are mentioned. This is irritating, because a non-significant increase should be considered as no change. For this purpose, we do statistics. Please remove these non-significant trends.
- Discussion length: Just like the Intro, the discussion is much too long. First of all, the writing style should be more compact and grammatically checked. Then, there is a lot of repetition of issues described in the Introduction. Please delete this unnecessary repetition. Furthermore, a number of less relevant issues is discussed and should be shortened/deleted. An example is the piece of text between lines 666 and 695. The authors did not even investigate insulin-stimulated (or AMPK-mediated) fatty acid transporter translocation. This is perhaps a missed opportunity, but given the role of the RabGAPs in basal transporter distribution, this omission may be justified. Another example is the extensive description of the changes in expression of the fatty acid transporters. This is a side finding and not directly related to TBCD1D1/TBC1D4 silencing. This section should be much shorter, and should end with a clear conclusion.
Other comments:
- Abstract: The term “functionally dominant” is out of place. This implies that the authors assigned a specific function to TBC1D1, and this function is then overruled by TBC1D4. This is not what is happening here. Specifically, TBC1D1 has no apparent function in fatty acid transporter dynamics. Please rephrase.
- Methods: In section 2.1, the 4 donors of adipose tissue biopsies are described as lean female patients undergoing laparoscopic cholecystectomy and in section 2.2 as healthy female donors. Please be consistent in the terminology.
- Methods: section 2,5: In total, 5 transfection groups were described, but only 4 groups appear in the figures. The appropriate control in Figs 4-10 should be the scrambled siRNA group (NC) rather than the non-transfected cells (Ctrl). The transfection procedure itself could already affect metabolic properties of the cells.
- Western blots in Figs. 4-8. Many “representative” Western blots do not reflect the corresponding quantification. This does not create confidence in the robustness of the data. A typical example is already the first blot in Fig. 4A (TBC1D1 expression in subADMSCs). Please provide better representative blots where needed.
- The PM fraction could perhaps still be used for Western detection of GLUT4. This would verify the established effects of TBC1D4 on GLUT4 translocation. Additionally, this would increase the confidence in the fractionation method.
- The terms “interestingly/importantly” and “seem to” are much too frequently used. Please remove all “seem to”s and reduce the number of “importantly”s.
- The authors refer in several places in the manuscript text to references (also from themselves) in which TBC1D4 deletion alters expression of fatty acid transporter in other cell types. What is the explanation? The role of RabGAPs in fatty acid transporter translocation is obvious, and is tightly linked to the regulation of Rab proteins driving the translocation process. But by which mechanism can RabGAPs alter expression levels of transporters?
- A clear explanation why TBC1D4 is more important in regulation of fatty acid uptake in subADMSCs compared to visADMSCs is lacking. And this should then be further discussed in the context of the different roles of both types of adipocytes in body lipid homeostasis.
Reviewer 2 Report
The authors aim at determining whether TBC1D1 and TBC1D4 play a role in lipid handling by adipocytes.
For that they isolated ADSMs from visceral and subcutaneous adipose tissues from 4 lean patients differentiated them into adipocytes. They down modulated the expression of TBC1D1 and TBC1D4 using transient transfection with siRNA and looked at the expression and the plasma membrane/vs cytosolic localization of several lipid transporters as well as palmitate uptake.
The experimental design is not well described: when the transfection is performed? Before or after the differentiation protocol. It seems it is before and thus the efficiency of differentiation must be precisely checked (lipid content/PPARg expression…). It is also unclear whether all the experiments have been performed with the ADSMs of the 4 patients. The adipocytes derived from vis ADMSCs exhibited no lipid droplets: are you sure they are differentiated?
The fractionation process is not experimentally verified: why transmembrane proteins are found in the cytosol? What are the proofs that the plasma membrane is only plasma membrane and that they are not contaminated by other intracellular membrane compartments (endosomes/Golgi/ER….). How to conclude about a change in the plasma membrane localization of a protein between the different experimental conditions when the loading control is also different.
The shown western blots are not always representative of the provided quantification: TBC1D1 expression is very low in the D4KD conditions in the western blot but not significantly changed in the quantification…
The western blots used as loading control in figure 5, figure 6, figure 7 for total protein amounts and cytosolic protein amounts are the same: it is not possible.
The effects on palmitate uptake are very small and it is difficult to believe they are significant. Insulin is regulating the activity of TBC1D4/TBC1D1. Why insulin has never been used in the experiments.
Reviewer 3 Report
In their manuscript Mikłosz et al. investigated the function of AS160/TBCD14 and TBC1D1 on fatty acid transport in adipose-derived mesenchymal stem cells (ADMSCs). Using siRNA they silenced the expression of AS160, TBC1D1 or both in ADMSCs from subcutaneous adipose tissue or visceral adipose tissue. Then, they study the total expression and membrane vs cytoplasmic expression of CD36, FABPpm, FATP1 and FATP4 and fatty acid uptake. From their results they propose that AS160 silencing induced an increase in palmitate uptake through an increase in membrane-localized CD36 and FABPpm.
One of the main problem of this manuscript, is that, in several places, what the authors write is not supported by the figures. Moreover, the effects are not reproducible between sub and visADMSCs or between single and double KO.
Figure 4:
In the western blot from fig 4C, in subADMSCs TBC1D1 expression is totally inhibited in D4KD cells (third line), however the graph indicate that its expression is not statistically modified.
The authors write (line 435-438) : “Additionally, we observed a tissue-specific pattern of Rab-GTPase-activating proteins, i.e. TBC1D1 expression was greater in the adipocytes from subADMSCs, whereas AS160/TBC1D4 protein expression was higher in the adipocytes derived from visADMSCs. “
To compare the expression between tissues, the different cell extracts should be analyzed, side by side, on the same Western blot.
Figure 5
-In fig 5D the quantification of the western blot shows an increase in plasma membrane CD36 in D4KD cells but the western blot provided below does not show such an increase.
- The authors write (lines 462-46) “membrane CD36/SR-B2 localization was increased upon TBC1D4 deficiency compared with the control cells (+64% and +72%, for D4KD and D1/D4KD vs Ctrl, p < 0.05, Figure 5D)” although this appears to be true for visADMSCS this is not the case for subADMSCs. Indeed, in D1KD/D4KD cells the plasma membrane of CD36 in subADMSC is not increased, compared to control.
Figure 6
In the Western blot (fig 6B), in subADMSCs, compared to the control condition, FABPm levels are unchanged in D1KD condition (lines 2) and decreased in D4KD. Why does the quantification shows the opposite?
The authors write (lines 485-486) “Comparing the two single knocked-down groups, we noticed an increased expression of FABPpm at the plasma membrane level for the D4KD group (+ 33% for D4KD vs D1KD, p < 0.05, Figure 6D).” This is true in subADMSC but not in visADMSCs , where a decrease is observed.
In the discussion the authors write (770-772): “We found that, in contrast to the TBC1D1-deficient adipocytes, there is a significant increase in plasmalemmal CD36/SR-B and FABPpm in TBC1D4 or double-deficient adipocytes”. However, Fig 6D shows that, compared to control, membrane associated FABPpm is not increased in TBC1D4 deficient cells (D4KD) but only in subADMSCs D1KD/D4KD (and not in visADMSCs D1KD/D4KD). So the increase in FABPpm membrane localization is observed only in the double KO and only in subADMSCs. No modification is observed in the single KO (D4KD) and not in the double KO in vis ADMSCs. These data argue against a function of FATPpm
Figure 10
Fig 10 shows that FFA, DAG and TAG contents are not modified after silencing of AS160 or TBC1D. The variations are, at best, around 10%, and are well within the error bars. Since there are no variations between the conditions the authors should not write: (Lines 606-607): “The total content of free fatty acids (FFA) remained relatively unchanged between all the experimental groups. Nevertheless, there was an overall trend towards greater total FFA concentration in the adipocytes originated from subcutaneous tissue (Figure 10).” And lines 613-617 “With regard to the total TAG content, no significant changes were observed between any of the experimental groups (Figure 10C, Table S4). Still, a tissue specific pattern seems to be discernible in the graph, i.e. greater total TAG content was found in the adipocytes originating from subcutaneous tissue independent of the gene silencing procedure (Figure 10C). “
In summary from these data it is not possible to draw any conclusion about the function of AS160 on fatty acid transport in adipocytes. In several places, the western blots provided do not match the quantifications. The effects observed are often not reproducible between sub and vis ADMSCs or between single and double KO. Moreover, this work is only correlative, trying to link modifications of expression of GTP-ase with localization of transporters. For instance, is the effect of AS160 on fatty acid uptake linked to a modification of localization of transporter or is it the result of its effect on glucose transport?
Others:
Figure 2 and 3 represent a classical protocol for purification and differentiation of ADMSCs. Since they do not provide new information they should be included in the supplementary data.
The introduction and the discussion would increase in clarity if they were significantly shortened and more focused.